# Efficacy of cochlear implants in children with borderline hearing who have already achieved significant language development with hearing aids

Young Seok Kim[1], Yehree Kim[1], Seung Jae Lee[1], Jin Hee Han[1], Nayoung Yi[1], Hyo Soon Yoo[1,2], Marge Carandang[3], Sang-Yeon Lee[2], Bong Jik Kim[4], Byung Yoon Choi[1,5]*

1 Department of Otorhinolaryngology, Seoul National University Bundang Hospital, Seongnam, South Korea, 2 Department of Otorhinolaryngology, Seoul National University Hospital, Seoul, South Korea, 3 Department of Otorhinolaryngology-Head and Neck Surgery, East Avenue Medical Center, Metro Manila, Philippines, 4 Department of Otorhinolaryngology, Chungnam National University Sejong Hospital, College of Medicine, Chungnam National University, Daejeon, Korea, 5 Sensory Organ Research Institute, Seoul National University Medical Research Center, Seoul, South Korea

* choiby2010@gmail.com

**Data Availability Statement:** All relevant data are within the paper and its Supporting information files.

## Abstract

There are still debates about timing and effectiveness of cochlear implants (CI) in pediatric subjects with significant residual hearing who do not belong to traditional indication of CI. In this study, we aimed to investigate the outcomes of CI, specifically on improvement of pronunciation, among hearing-impaired children already with a substantial degree of language skills as evaluated by Categories of Auditory Perception (CAP) scores or sentence score. Our cohort comprised pediatric CI recipients from July 2018 through October 2020. Among them, cases with CAP scores of 5 or 6 preoperatively were defined as "borderline cases". We investigated prevalence and etiologies, and compared speech evaluation data preoperatively and postoperatively at three time points (3, 6 and 9–12 months after implantation). Among 86 pediatric CI recipients, 13 subjects (15.12%) had language development that reached CAP scores of 5 or 6 before implantation. Postoperative speech evaluation data 6 months after implantation revealed significant improvement of pronunciation (Urimal Test of Articulation and Phonation scores: UTAP), Infant-Toddler Meaningful Auditory Integration Scale (IT-MAIS) and word perception scores, but not of CAP and sentence perception scores. Notably, the significant improvement of pronunciation based on UTAP scores outstripped that of other speech parameters and this continued steadily up to one-year postoperatively. The result of the study serves as evidence for what to expect from cochlear implantation in hearing-impaired children who have already achieved a substantial degree of language development in terms of CAP scores or sentence perception scores, preoperatively.

**Funding:** This study was supported by the Basic Science Research Program through the National Research Foundation of Korea (NRF) funded by the Ministry of Education (2018R1A2B2001054 to B.Y. C.) and the research funds of Seoul National University Bundang Hospital (16-2019-006 to B.Y. C) (13-2018-015 to B.Y.C.). There was no additional external funding received for this study. All the funders and sources of support had no role in study design, data collection and analysis, decision to publish, or preparation of the manuscript.

**Competing interests:** The authors have declared that no competing interests exist.

## Introduction

The need for hearing rehabilitation through early cochlear implantation (CI) in children with bilateral profound sensorineural hearing loss (SNHL) is already established. With the proven efficacy and development of CI technology, subjects who had not been previously operated are becoming more and more adaptable to CI [1]. However, there are still controversies about timing and effectiveness of CI in pediatric subjects who have already achieved significant language development with hearing aids. The controversies basically come from difficulties in anticipating how much more CI can facilitate language development, including expressive language, among pediatric subjects with significant residual hearing who has already achieved open set word and sentence recognition using hearing aids, but not in the same level as their peers without hearing impairments. There has been scarcity of reports addressing this issue.

Although there have been several reports dealing with hearing benefits after CI in pediatric subjects with significant residual hearing [2, 3], these did not clearly address the benefits of CI in terms of the improvement of various language skills, especially expressive components, among children in whom receptive language development has already been significantly made. Further, with the growing importance of sophisticated pronunciation/consonant production in social life, there is increasing demand for evaluation of the benefit of CI on pronunciation in hearing-impaired children who already have sufficient receptive language skills through hearing aids but lacks a sophisticated consonant production.

In this study, we aimed to investigate the proportion and etiologies of pediatric CI candidates already with a substantial degree of open set language skills obtained using hearing aids, and to address short-term outcomes of CI among such pediatric subjects, focusing on improvement of pronunciation.

## Materials and methods

### Study subjects

We prospectively established a pediatric CI cohort comprising 112 children under 15 years of age who underwent unilateral or bilateral CI at Seoul National University Bundang Hospital (Seongnam, Korea) by a single surgeon (B.Y.C.) from July 2018 through October 2020. Revision or secondary in sequential CI cases (n = 26, (23%)) were disqualified. Among the remaining 86 pediatric implantees, we focused on a subset of cases who preoperatively already obtained Categories of Auditory Perception (CAP) scores of 5 or 6 indicative of acquisition of open set sentence recognition to some extent. We named this subcohort as "borderline cases". Unilateral cochlear implantation was performed in the worse ear in these borderline cases, with a few exceptions, not taking the risk of damaging residual hearing of the better ear. In order to clarify the safety of the operation, the hearing preservation rate at postoperative 3 months was calculated among the implanted ears with significant residual hearing (pure tone thresholds ≤85dB HL at 250 and 500 Hz) preoperatively, based on classification suggested by HEARRING group [4].

To determine the characteristics of these borderline cases, diagnosed etiologies and audiologic test results were collected. We compared preoperative speech evaluation data and at 3 months-, 6 months- and 9–12 months-postoperatively. These included CAP scores, Infant-Toddler Meaningful Auditory Integration Scale (IT-MAIS), Speech Intelligibility Rating (SIR), perception scores (%) of Spondee word and sentence recognition test using Korean version of the Central Institute for the Deaf (K-CID), and Urimal Test of Articulation and Phonation (U-TAP) scores. In cases with multiple preoperative speech evaluation data, data from the last evaluations before implantations were used. Available speech evaluation data at 3 to 6 months

and 12 to 15 months before implantation were also collected to assess the effect of hearing aids before CI. Subjects without proper speech evaluation data were excluded from analysis.

Before molecular genetic testing, we obtained the written consents from the subjects and/or their parents about genetic testing itself and possibility of publishing genetic test results. However, written consent was exempted for review and publication of their clinical information because this retrospective study design did not do any harm to patients. These consent procedures and the study was approved by the Seoul National University Bundang Hospital Institutional Review Board (IRB Number B-2107-696-105).

## Etiologic diagnosis of borderline cases

According to our etiologic diagnostic pipeline for pediatric CI candidates with SNHL, all available methods were used including internal auditory canal magnetic resonance imaging (IAC-MRI) and Molecular genetic testing (MGT). Except for subjects with already known etiologies, MGT was conducted with the following process: [1] U-Top screening kit [5, 6] or Direct Sanger sequencing in subjects with a characteristic phenotype, such as enlarged vestibular aqueduct, [2] deafness panel sequencing (TES-129) [7, 8] or exome sequencing [9, 10], and [3] fluorescent in situ hybridization (FISH), if necessary [11]. Panel sequencing or exome sequencing results were analyzed as previously described [12, 13].

## Speech evaluation

CAP scores, IT-MAIS scores and perception scores of Spondee word and K-CID were collected. The CAP scores and IT-MAIS scores were collected based on the answers given by caregivers. The CAP score lies on an eight-point scale ranging from no awareness of environmental sounds (category 0) to conversational use of the telephone with a known speaker (category 7) [14]. IT-MAIS score was calculated with the instrument including 10 items in three domains: vocalization behavior (items 1–2), alerting to sounds (items 3–6), and deriving meanings from sounds (items 7–10). Each item is scored from 0 to 4: 0, never; 1, rarely (25%); 2, occasionally (50%); 3, frequently (75%); and 4, always (100%) [15]. The perception scores were evaluated by an experienced speech-therapist with Spondee words and K-CID at 70 dB SPL using the subject's hearing aids in audio-only conditions. The K-CID was composed of speech sentences in everyday situations. The subjects repeat the sentences containing keywords verbally and the scores are calculated as the ratio of correctly identified keywords (%).

The SIR is a measurement of speech production in real life. It categorizes the degree of intelligibility of the subject's speech from 1 (unintelligible, prerecognizable words) to 5 (intelligible to all listeners) [16]. It was recorded by the speech-therapist during speech evaluation sessions.

U-TAP test was used for evaluation of pronunciation. It is an articulation and phonation test for children widely used in Korea [17]. It tests consonant and vowel accuracy in word level and sentence level. During the tests, subjects are told to produce words with 30 colored pictures and encouraged to speak 1–3 sentences containing keywords while looking at 9 pictures. The examiner measures the accuracy rates of 43 consonants and 10 vowels. U-TAP scores (%) were measured as accuracy rates of consonants.

## Statistical analyses

Considering a small number of the subjects, a nonparametric statistical method was used to compare preoperative speech evaluation data and at 3-, 6-, and 9–12 months postoperatively. Wilcoxon signed rank test was used to compare CAP, IT-MAIS, SIR, perception scores and U-TAP scores within the subjects. A p-value<0.05 was considered to be statistically significant.

All analyses were performed using the R software package, version 3.3.2 (R Foundation for Statistical Computing, Vienna, Austria).

## Results and discussion

### Study participants

Among 86 pediatric CI recipients, 13 subjects (13/86 = 15.12% of pediatric CI recipients for their first time) had obtained borderline language development that reached CAP scores of 5 or 6 just prior to or at the time of cochlear implantation (Table 1). Overall, 16 ears from these 13 subjects were implanted; one subject (subject 2) underwent bilateral simultaneous CI and two subjects (subject 1 and 8) had sequential bilateral CI with an interval of 4 to 5 months. In the other 10 subjects, unilateral cochlear implantation was performed in the worse ear, not taking the risk of damaging residual hearing of the better ear. The age range was 26–115 months (median 67 months) at the time of implantation. There was no pediatric subject with surgical complication during or after implantation.

In our cohort, 12 ears from 9 subjects turned out to have significant residual hearing (pure tone thresholds ≤85dB HL at 250 and 500 Hz) preoperatively. Hearing preservation rates at 3 months postoperatively, based on classification suggested by HEARRING group [4] (S1 and S2 Tables), from nine (75.0%) of the 12 ears fell into the criteria of either complete (>75%) or partial (25–75%) preservation.

The mean pure tone averages (500, 1000 and 2000 Hz) of 13 subjects were 73dB (right side) and 71dB (left side). Click auditory brainstem response (ABR) threshold averages were 79/71dB (right/left) when no response was calculated as 100dB. Among 13 subjects, 4 subjects had asymmetric hearing loss (No. 6 and 11–13) and one subject (No. 10) was a single sided deafness case.

### Etiologic diagnosis of borderline cases

Among these 13 borderline cases, we were able to make the etiologic diagnosis from 12 subjects (92.3%) using IAC-MRI and MGT (S3 Table). Congenital cytomegalovirus (CMV)

**Table 1. Characteristics of subjects included.**

| No. | Age at CI (months) | Sex | Etiology | Pure tone threshold (R/L, dB HL) | WRS (R/L, %) | ABRT (R/L, dB nHL) | CI Side | CI model |
|---|---|---|---|---|---|---|---|---|
| 1 | 54 | M | USH2A | 78/73 | 20/24 | NR/NR | R -> L | CI 522 |
| 2 | 63 | F | PDZD7 | 70/72 | 8/8 | 75/70 | B | CI 522 |
| 3 | 67 | F | SLC26A4 | 43/63 | 72/48 | 70/70 | L | CI 532 |
| 4 | 70 | M | MYO15A | 70/68 | 48/48 | 90/90 | L | CI 532 |
| 5 | 94 | M | CMV most likely | 77/83 | 72/68 | 70/80 | L | CI 532 |
| 6 | 71 | F | CMV most likely | 47/113 | 80/0 | 70/100 | L | CI 532 |
| 7 | 74 | F | MYO6A | 78/83 | 44/40 | 55/55 | L | CI 532 |
| 8 | 115 | F | Unknown | 75/97 | 40/16 | 90/NR | L -> R | CI 532 |
| 9 | 48 | M | SLC26A4 | 95/80 | 0/36 | NR/80 | R | CI 632 |
| 10 | 67 | M | NLRP3 | 33/87 | 96/8 | 40/85 | L | CI 632 |
| 11 | 48 | F | CMV most likely | 110/50 | CNT | NR/45 | R | CI 532 |
| 12 | 26 | M | CMV most likely | 100/10 | CNT | NR/20 | R | CI 532 |
| 13 | 26 | M | SLC26A4 | 70/40 | CNT | 70/35 | R | CI 632 |

Pure tone threshold denotes the average pure tone thresholds at 0.5, 1, and 2 kHz without hearing aids. WRS was measured at 30dB higher than speech recognition test (SRT) or at 100dB. CI, cochlear implant; R, right side; B, both sides; L, left side; WRS, word recognition score; ABRT, auditory brainstem response threshold; M, male; F, female; NR, no response; CMV, cytomegalovirus; CNT, cannot be examined.

infection was diagnosed or highly suspected in four subjects, based on IAC-MRI, audiological phenotypes, CMV viral culture and CMV-PCR testing.

Among 8 subjects whose etiologic diagnosis was made based on MGT, *SLC26A4* variants were the most common cause (3/8, 37.5%) and there were one each of cases with variants in *USH2A, PDZD7, MYO15A, MYO6* and *NLRP3* genes.

### Hearing rehabilitation with hearing aids before cochlear implantation

Among the 13 borderline subjects, there were four subjects with available speech evaluation data at 3 to 6 months prior to surgery, and there was only one subject with speech evaluation data as early as 15 to 18 months prior to CI. To assess the effects of hearing rehabilitation with conventional hearing aids, U-TAP scores before implantations (3 to 6 prior to surgery vs. just before surgery) were compared (Fig 1). For these four subjects, there was no improvement of pronunciation even with rigorous speech therapy using conventional hearing aids for 3–6 months (U-TAP score, 80.8±13.1 to 81.3±13.7, p = 1, Wilcoxon signed rank test), prompting CI in these subjects. At the time of decision of CI, SIR and UTAP scores reflecting speech intelligibility/pronunciation, as well as Spondee word perception scores, showed a wide distribution among these nine subjects (Fig 2).

### Results of cochlear implantation

Among these 13 subjects, 4 subjects (Pt. 10–13) were excluded from post cochlear implantation outcome analysis. Pt 10 was diagnosed as CINCA syndrome associated with NLRP3

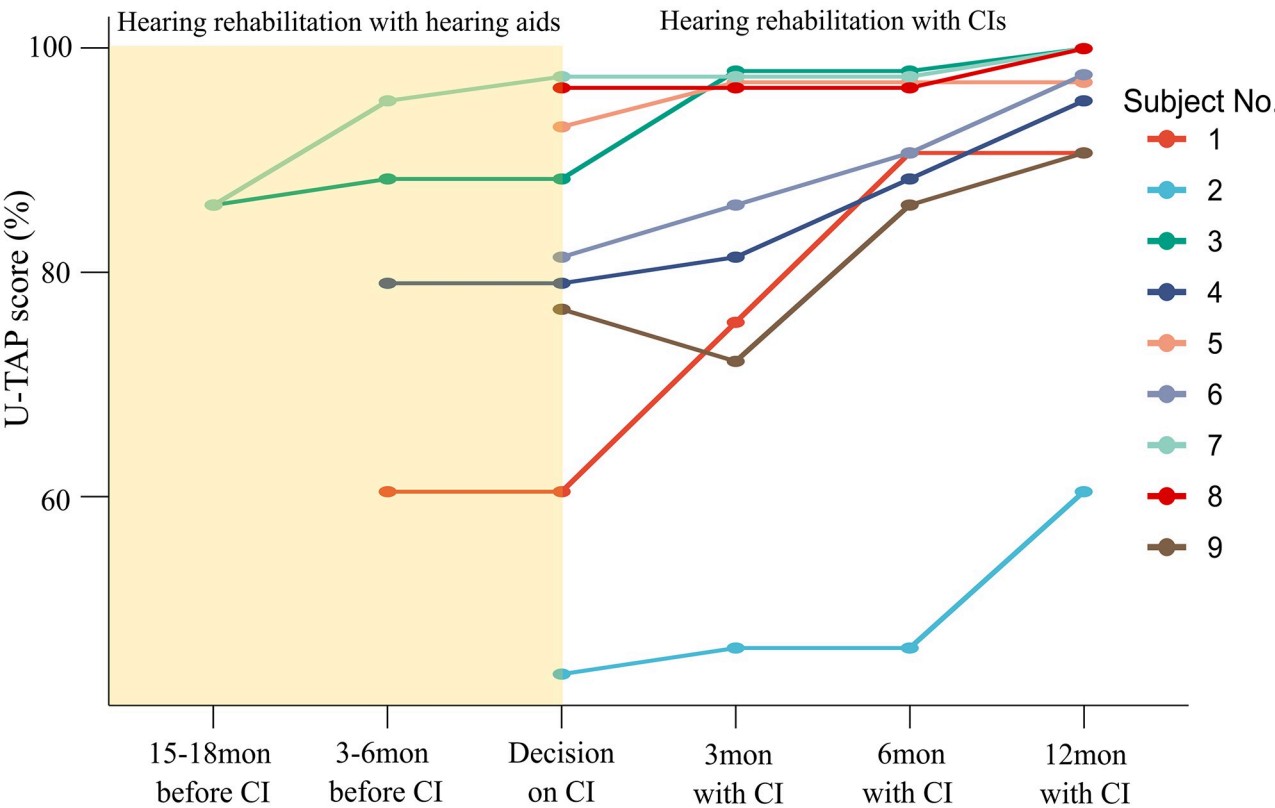

**Fig 1. Comparisons of U-TAP scores with hearing aids before implantation and with cochlear implants in child cochlear implantees with borderline language development.** U-TAP, Urimal Test of articulation and Phonation; CI, cochlear implant.

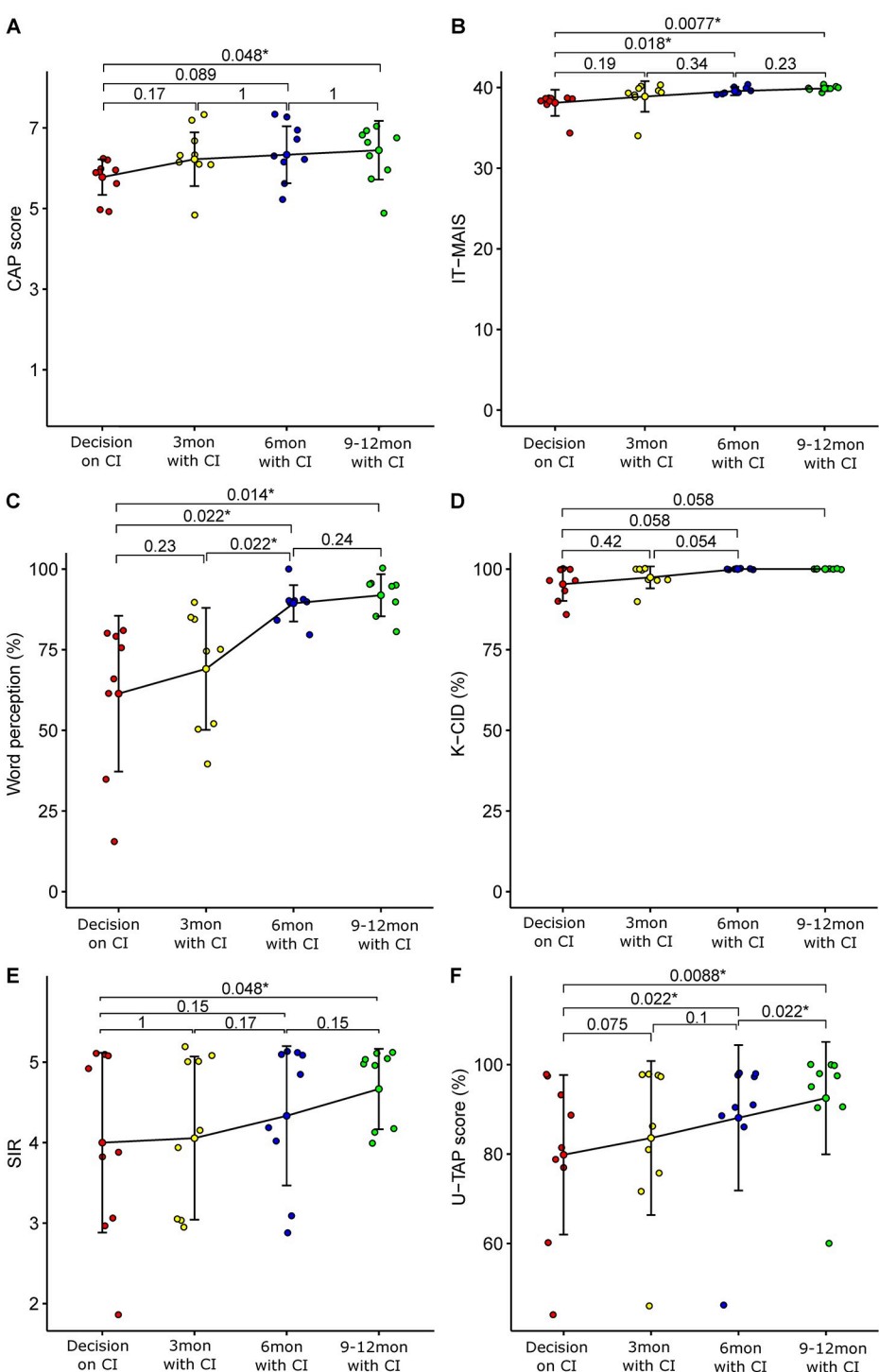

**Fig 2. Comparisons of speech evaluation results between preoperative and post-implantation in child cochlear implantees with borderline language development.** A, CAP. B, IT-MAIS. C, SIR. D, Spondee word perception. E, sentence perception in K-CID. F, U-TAP. The p-values are notated above the comparison lines (Wilcoxon signed rank test). CI, cochlear implant; CAP, Categories of Auditory Performance; IT-MAIS, Infant-Toddler Meaningful Auditory Integration Scale; SIR, Speech Intelligibility Rating; K-CID, Korean version of the Central Institute for the Deaf; U-TAP, Urimal Test of articulation and Phonation.

mutation and showed features of single sided deafness (SSD). This subject showed great improvement of U-TAP score, Word perception and K-CID, but was excluded from statistical analysis because it could undermine the unity of our cohort (S4 Table). Pt 11 and Pt 12 were congenital CMV infection and they could not perform several tests during speech evaluation due to their cognitive impairment. Pt. 13 had not performed post-CI speech evaluation at post-operative 3 and 6 months, and postoperative 9 months evaluation was conducted at our branch hospital, which made comparing data unreliable. In total, 9 subjects (Pt. 1–9) were included in post-CI outcome analysis (S4 Table, Fig 2). The overall auditory receptive abilities were indicated by CAP and IT-MAIS; speech perception was shown by Spondee word perception and K-CID sentence perception; speech intelligibility/pronunciation were measured by SIR and U-TAP scores.

IT-MAIS (38.1±1.5 to 39.6±0.5, p = 0.018, Wilcoxon signed rank test), Spondee word perception (61.4±22.6 to 87.2±7.9, p = 0.022, Wilcoxon signed rank test) and U-TAP scores (mean score; from 79.84±16.81 to 88.11±15.33, p = 0.022, Wilcoxon signed rank test) showed significant improvement at 6 months after implantation (Fig 2). Most importantly, U-TAP scores continued to show statistically significant improvement from 6 months to 12 months after CI (mean score; from 88.11±15.33 to 92.50±11.85, p = 0.022, Wilcoxon signed rank test) (Fig 2). This improvement sharply contrasted with the UTAP scores which were stagnant for 3–6 months prior to cochlear implantation (Fig 1). This was well shown especially in the U-TAP scores of subject No. 1, 3 and 4 (Fig 1). In summary, pronunciation, which became stagnant and didn't improve further with hearing aids in these borderline cases, improved significantly and consistently with cochlear implantation throughout 12 months after cochlear implantation.

In contrast, we did not obtain significant improvement of CAP scores (5.8±0.4 to 6.3±0.66, p = 0.089, Wilcoxon signed rank test), K-CID sentence scores (95.3±4.9 to 100, p = 0.058, Wilcoxon signed rank test) and SIR (4.0±1.0 to 4.3±0.8, p = 0.15, Wilcoxon signed rank test) at 6 months after implantation. It was not until 9 to 12 months after CI that overall improvement of CAP scores and SIR reached significant level (5.8±0.4 to 6.4±0.7, p = 0.048, Wilcoxon signed rank test).

## Discussion

In our cohort, 15.12% (13/86) of pediatric CI recipients already had CAP of 5 or 6 before their first CI surgery. Likewise, the preoperative K-CID scores already reached 95% and in terms of speech production skills measured with SIR. However, there had been no significant improvement of pronunciation, as reflected by stagnant preoperative U-TAP scores with an average 80.8% at the time of cochlear implantation. Indeed, significant improvement of U-TAP scores as well as IT-MAIS and word perception were observed during the first 6 months of auditory rehabilitation using cochlear implants. U-TAP scores also showed further significant improvement from 6 months to 12 months after cochlear implantation. At 9 to 12 months after cochlear implantation, we were still able to observe significant improvement of CAP and SIR scores even though these were already nearly a full mark preoperatively. These results suggest that: first, in children with moderately severe or severe hearing loss who have already achieved a substantial degree of receptive language development, the final task in language development is speech intelligibility/pronunciation; and second, cochlear implantation can still offer better audiological benefit to these children especially in terms of speech intelligibility/pronunciation.

Some of our results about word recognition/discrimination ability were in line with those in the literature. Recently, Na et el. reported improvement of speech perception after CI in children with preoperative residual hearing (pure tone audiometry thresholds ≤90dB HL)

[18]. In another study with 53 pediatric recipients with significant residual hearing, significant improvement of word discrimination were noted at 12- and 24-months after CI [2]. Remarkably, children <10 years at time of operation showed significantly better word discrimination outcomes than older children, suggesting that there is a more effective surgical time window for efficient speech perception development. Park et al. reported that delaying CI was significantly correlated with poorer word recognitions in pediatric candidates with progressive hearing loss and substantial residual hearing (pure tone threshold average ≤75dB HL) at the time of implantation [19]. Improvement from these borderline cases was not limited to just word discrimination/recognition ability but extended to expressive language or speech production/ intelligibility. Indeed, Matthew et al. reported significant improvement of speech recognition and facilitated receptive and expressive language development in pediatric CI recipients with less severe hearing loss and recognition scores already greater than 30% [20]. Further, Wilson et al. reported meaningful improvement in CAP and even SIR scores from 23 pediatric CI recipients with significant preoperative and postoperative residual hearing [3]. To the best of our knowledge, however, no papers have rigorously analyzed the effect of CI on higher level of language development including improvement of pronunciation in pediatric subjects who already have achieved a significant degree of receptive language development using conventional hearing aid. Due to methodological differences, the effect of CI on the child's pronunciation development has been less clearly elucidated [21]. In our cohort, all subjects showed improvement of U-TAP scores and three of nine (33.3%) reached the perfect score after cochlear implantation. Our study merits attention for it observed rapid and sustained improvement of pronunciation throughout full one year postoperatively from those patients who previously stagnated for 3–6 months using conventional hearing aids.

There is no clear audiologic cut-off point for CI candidacy, resulting in variable selection criteria proposed by institutions [22–24]. Given the difficulty of audiologic evaluation and its relevance to language development, decision making on CI in pediatric subjects tends to be more complicated and important than in adults [25]. Hearing thresholds are important for CI candidacy, but the progress on speech perception and auditory development questionnaires should also be considered as primary determinants [20]. In this regard, our study selected the target subjects based on the CAP score. Children who are able to understand common phases without lip reading (CAP score 5 and above) may not have been considered as traditional canonical candidates for CI, however, they still had significant audiological benefits at 6 months after cochlear implantation in our cohort.

Of course, even without cochlear implantation, these subjects may possibly have exhibited a certain amount of improvement in IT-MAIS, Spondee word perception or U-TAP, questioning the legitimacy of CI in these borderline cases. However, this is least likely the case for U-TAP, considering there was no improvement of U-TAP score within 3–6 months prior to cochlear implantation.

Preservation of a substantial portion of residual hearing has been reported to be more feasible in pediatric patients than in adults [26, 27]. Our cohort also showed reasonable range of hearing preservation, (9/12 = 75.0%) with either complete or partial preservation at least in short-term follow up, significantly alleviating concern of parents related to potential loss of the residual hearing of their kids. Further, the etiology of a substantial portion of our cohort was related to either *SLC26A4* variants (3 subjects) or suspected congenital CMV infection (4 subjects), both of which are common causes of progressive hearing loss in infants [28, 29]. With these etiologies, it is highly probable that the hearing status of both worse and better ears will deteriorate over time. Therefore, if an important parameter such as UTAP scores in speech development falls into a stagnant phase using conventional hearing aids, CI can be considered as a reasonable option. In particular, development of consonant production/pronunciation in

these children is faster while they are young [30, 31] and development of speech production after cochlear implantation in prelingually deaf children seems to approach a plateau after a certain period [32]. Therefore, sticking to rehabilitation with hearing aids even after reaching the age in children where improvement of consonant production would plateau due to limited amplification could pose significant limitations.

In contrast to the significant improvement in word perception scores and consonant production after CI, the changes in sentence perception scores (K-CID) were not statistically significant. Preoperatively, 7 out of 8 subjects had already achieved a score of 90 or higher even before the implantation, yielding a ceiling effect. Thus, in borderline cases, it seems that determining the indications for CI only by relying on dB and K-CID scores is inadequate. It is also necessary to thoroughly consider word perception and, more importantly, consonant accuracy and pronunciation. However, to the best of our knowledge, there is no study following the development of consonant production/pronunciation in borderline cases like ours. Development of expressive language, including pronunciation, requires more rehabilitation period than development of receptive language. In the typical language development course, receptive language development precedes expressive skills [33, 34]. Language development of pediatric cochlear implantees was reported to follow normative developmental trajectories [35, 36]. Spencer et al. reported consonant acquisition patterns of child cochlear implantees were also similar to peer groups with normal hearing [21]. In line with this, the previous study reported steady improvement of phoneme accuracy for 6 years after cochlear implantations in 27 pediatric recipients [32].

Although our study observed the development in a wide range of language domains, limitations of the study include: small number of subjects; relatively short-term follow-up period; and lack of control group, because it is hard to observe and withhold surgery when the need for cochlear implant is discovered. To overcome this limitation, we also demonstrated that no significant improvement of pronunciation had been achieved with hearing aids before CI in cases with available data. As the indications for CI are expanding, additional studies using a larger cohort are expected to reveal more information.

## Conclusions

In pediatric subjects with borderline hearing loss who already had CAP of 5 or 6 and 90% or more of KCID scores before surgery, CAP and SIR scores were greatly affected by ceiling effects do not show any significant improvement until 9–12 months. However, statistically significant improvement of scores in IT-MAIS, Spondee word perception scores and U-TAP were observed as early as 6 months postoperatively. It is important to note that the improvement of pronunciation as reflected by UTAP outstrips that of other speech evaluation parameters and continued steadily upto one-year postoperatively. The results of the study are expected to serve as evidence for deciding whether to have CI surgery and level expectations on cochlear implantation in children who have already achieved a substantial degree of language development based on CAP scores or sentence score.

## Supporting information

**S1 Table. Residual hearing preservation of pediatric cochlear implantees with borderline receptive language developments at 3 months after implantation.** Hearing preservation was calculated in subjects with functional residual hearing (pure tone thresholds ≤85dB HL at 250 and 500 Hz), RH, residual hearing.
(DOCX)

**S2 Table. Audiometry results at preoperative, preoperative with hearing aids and 3, 6 and 9 to 12 months after cochlear implant.** In several subjects without hearing aids used, hearing tests with rental hearing aids were performed for preoperative evaluation. Patients who underwent play audiometry due to their young age were missed at some frequencies. CI, cochlear implants; HA, hearing aids; Preop., at preoperative; Pre-HA, preoperative with hearing aids; 3M, at postoperative 3 months; 6M, at postoperative 6 months; 9-12M, at postoperative 9 to 12 months.
(DOCX)

**S3 Table. Details of genotypes of pediatric cochlear implantees with borderline receptive language developments before surgery.** Het, heterozygote; Homo, homozygote; VUS, Variant of Uncertain Significance; NA, not applicable; ND, no data.
(DOCX)

**S4 Table. Speech evaluation results at preoperative and 3, 6 and 9 to 12 months after cochlear implant.** U-TAP, Urimal Test of Articulation and Phonation; SD, standard deviation; K-CID, Korean version of the Central Institute for the Deaf; Preop., at preoperative; 3M, at postoperative 3 months; 6M, at postoperative 6 months; 9-12M, at postoperative 9 to 12 months; UC, uncheckable.
(DOCX)

## Acknowledgments

The authors declare that we have no competing interests. These data have not been previously published and are not submitted elsewhere for publication.

## Author Contributions

**Conceptualization:** Young Seok Kim, Byung Yoon Choi.

**Data curation:** Young Seok Kim.

**Formal analysis:** Young Seok Kim.

**Investigation:** Young Seok Kim, Yehree Kim, Seung Jae Lee, Byung Yoon Choi.

**Methodology:** Byung Yoon Choi.

**Resources:** Seung Jae Lee, Jin Hee Han, Nayoung Yi, Hyo Soon Yoo, Sang-Yeon Lee, Bong Jik Kim, Byung Yoon Choi.

**Visualization:** Young Seok Kim.

**Writing – original draft:** Young Seok Kim, Byung Yoon Choi.

**Writing – review & editing:** Yehree Kim, Marge Carandang.

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
