## [Decision Letter · Decision Letter 0]

25 Jan 2022

PONE-D-21-38274Efficacy of cochlear implants in children with borderline hearing who have already achieved significant language development with hearing aidsPLOS ONE

Dear Dr. Byung Yoon Choi,

Thank you for submitting your manuscript to PLOS ONE. After careful consideration, we feel that it has merit but does not fully meet PLOS ONE’s publication criteria as it currently stands. Therefore, we invite you to submit a revised version of the manuscript that addresses the points raised during the review process.

We look forward to receiving your revised manuscript.

Kind regards,

Shin-ichi Usami, M.D., Ph.D.

Academic Editor

PLOS ONE

Journal Requirements:

a) Did participants provide their written or verbal informed consent to participate in this study?

3.  You indicated that you had ethical approval for your study. In your Methods section, please ensure you have also stated whether you obtained consent from parents or guardians of the minors included in the study or whether the research ethics committee or IRB specifically waived the need for their consent.

( This study was supported by the Basic Science Research Program through the National Research Foundation of Korea (NRF) funded by the Ministry of Education (2018R1A2B2001054 to B.Y.C.) and the research funds of Seoul National University Bundang Hospital (16-2019-006 to B.Y.C) (13-2018-015 to B.Y.C.). The funders had no role in study design, data collection and analysis, decision to publish, or preparation of the manuscript.)

Additional Editor Comments:

The reviewer has noted a substantial number of points that require further attention. Therefore, I invite you to respond to the reviewers' comments and to revise your manuscript accordingly. Please note that final acceptance of your manuscript is conditional upon you responding adequately and appropriately to the comments and criticisms of the reviewer.

Reviewers' comments:

Reviewer's Responses to Questions

**Comments to the Author**

1. Is the manuscript technically sound, and do the data support the conclusions?

Reviewer #1: Partly

2. Has the statistical analysis been performed appropriately and rigorously? 

Reviewer #1: Yes

3. Have the authors made all data underlying the findings in their manuscript fully available?

Reviewer #1: No

4. Is the manuscript presented in an intelligible fashion and written in standard English?

Reviewer #1: Yes

5. Review Comments to the Author

Reviewer #1: This manuscript deals with the postoperative outcome for those with limited progress made with conventional hearing aid use with cochlear implantation. As the authors pointed out, there are uncertainties regarding the point of surgical intervention using cochlear implantation especially when a child has residual hearing. Thus, the authors are addressing clinically relevant and important questions. Although the stated questions are valid , there are some areas that require further clarification.

There are total of 13 patients who were selected for this current study but it is unclear as to when the patients started using hearing aids for the first time, and how long the hearing aids were used prior to surgery. It appears that some of the patients in the study did not have any usage of hearing aid. Patient 2 can be defined as a case of delayed intervention with bilateral congenital profound hearing loss without any prior use of hearing aids. By reviewing the audiometric criteria presented for the 13 patients, some of the patients can be characterized as receiving cochlear implantation for single sided deafness. Unless the patient selection criteria is better defined, there are some selection bias. The duration of deafness, clear audiologic definition for surgical intervention, methods of preoperative and postoperative rehabilitations are not available. Thus, the authors are encouraged to present detailed clinical information for each patient. In this study, the definition of functional residual hearing threshold was defined as 85dB but, it is highly questionable that a patient can obtain speech discrimination at this level using a hearing aid. Therefore, the definition of functional hearing preservation needs better definition.

In the supplement material provided, the hearing preservation data is presented only at three months following surgery only qualitative data is presented. Since these patients were followed for 12 months following surgery, it is recommended to present the actual audiometric data for these nine patients. Since the number of patients in the current study is rather small, the authors are encouraged to present all of the longitudinal audiologic and speech perception data in the first 12 months period at the individual level. Once all of the audiologic data and clinical data are available for all of these patients, authors should also look into the difference of outcome using different cochlear implant electrodes.

For the revision, it is recommended to shorten the manuscript since there are some areas that are redundant.

6. PLOS authors have the option to publish the peer review history of their article (what does this mean?). If published, this will include your full peer review and any attached files.

Reviewer #1: No

---

## [Author Response · Author response to Decision Letter 0]

13 Feb 2022

Reviewer #1: This manuscript deals with the postoperative outcome for those with limited progress made with conventional hearing aid use with cochlear implantation. As the authors pointed out, there are uncertainties regarding the point of surgical intervention using cochlear implantation especially when a child has residual hearing. Thus, the authors are addressing clinically relevant and important questions. Although the stated questions are valid, there are some areas that require further clarification.

There are total of 13 patients who were selected for this current study but it is unclear as to when the patients started using hearing aids for the first time, and how long the hearing aids were used prior to surgery. It appears that some of the patients in the study did not have any usage of hearing aid. Patient 2 can be defined as a case of delayed intervention with bilateral congenital profound hearing loss without any prior use of hearing aids. By reviewing the audiometric criteria presented for the 13 patients, some of the patients can be characterized as receiving cochlear implantation for single sided deafness. Unless the patient selection criteria is better defined, there are some selection bias. The duration of deafness, clear audiologic definition for surgical intervention, methods of preoperative and postoperative rehabilitations are not available. Thus, the authors are encouraged to present detailed clinical information for each patient. In this study, the definition of functional residual hearing threshold was defined as 85dB but, it is highly questionable that a patient can obtain speech discrimination at this level using a hearing aid. Therefore, the definition of functional hearing preservation needs better definition.

-> Thank you for the opportunity to supplement the important points. We added the data of timing and sides of hearing aids and preoperative and postoperative audiometry of each patients in supplementary table S2. In these borderline cases, we decided cochlear implantation not by audiologic definition but by consultation with the guardian in consideration of language development including pronunciation accuracy. 

The reason we dealt with the preservation of residual hearing in this study was to clarify the safety of the surgery in these subjects. So pure tone thresholds ≤85dB HL at 250 and 500 Hz are the thresholds used as a standard for measuring rates of preserving residual hearing in cochlear implant surgery suggested by HEARRING group. To make the point clear, we edited the manuscript. Please refer line 75-78 and 285-288 of the manuscript. 

In the supplement material provided, the hearing preservation data is presented only at three months following surgery only qualitative data is presented. Since these patients were followed for 12 months following surgery, it is recommended to present the actual audiometric data for these nine patients. Since the number of patients in the current study is rather small, the authors are encouraged to present all of the longitudinal audiologic and speech perception data in the first 12 months period at the individual level. Once all of the audiologic data and clinical data are available for all of these patients, authors should also look into the difference of outcome using different cochlear implant electrodes.

-> Thanks for the good point. Audiologic and speech evaluation data including CAP, IT-MAIS and SIR scores for 12 months in each subjects were added in the supplementary material. As shown in Table 1 of main manuscript, CI 522, 532 and 632 electrodes were used. Because of small number of subjects, we could not study the difference between different electrodes. 

For the revision, it is recommended to shorten the manuscript since there are some areas that are redundant.

-> Thanks for the good comment. The manuscript has been edited to be more concise.

---

## [Decision Letter · Decision Letter 1]

19 Apr 2022

Efficacy of cochlear implants in children with borderline hearing who have already achieved significant language development with hearing aids

PONE-D-21-38274R1

Dear Dr. Byung Yoon Choi,

We’re pleased to inform you that your manuscript has been judged scientifically suitable for publication and will be formally accepted for publication once it meets all outstanding technical requirements.

Kind regards,

Shin-ichi Usami, M.D., Ph.D.

Academic Editor

PLOS ONE

Additional Editor Comments (optional):

Reviewers' comments:

Reviewer's Responses to Questions

**Comments to the Author**

1. If the authors have adequately addressed your comments raised in a previous round of review and you feel that this manuscript is now acceptable for publication, you may indicate that here to bypass the “Comments to the Author” section, enter your conflict of interest statement in the “Confidential to Editor” section, and submit your "Accept" recommendation.

Reviewer #1: All comments have been addressed

2. Is the manuscript technically sound, and do the data support the conclusions?

Reviewer #1: Yes

3. Has the statistical analysis been performed appropriately and rigorously? 

Reviewer #1: Yes

4. Have the authors made all data underlying the findings in their manuscript fully available?

Reviewer #1: Yes

5. Is the manuscript presented in an intelligible fashion and written in standard English?

Reviewer #1: Yes

6. Review Comments to the Author

Reviewer #1: All of the questions have been adequately addressed. Therefore, I recommend the submitted manuscript to be accepted.

7. PLOS authors have the option to publish the peer review history of their article (what does this mean?). If published, this will include your full peer review and any attached files.

Reviewer #1: No

---

## [Editor Report · Acceptance letter]

23 May 2022

PONE-D-21-38274R1 

Efficacy of cochlear implants in children with borderline hearing who have already achieved significant language development with hearing aids 

Dear Dr. Choi:

I'm pleased to inform you that your manuscript has been deemed suitable for publication in PLOS ONE. Congratulations! Your manuscript is now with our production department. 

Kind regards, 

on behalf of

Dr. Shin-ichi Usami 

Academic Editor

PLOS ONE